# Focal Boost in Prostate Cancer Radiotherapy: A Review of Planning Studies and Clinical Trials

**DOI:** 10.3390/cancers15194888

**Published:** 2023-10-08

**Authors:** Yutong Zhao, Annette Haworth, Pejman Rowshanfarzad, Martin A. Ebert

**Affiliations:** 1School of Physics, Mathematics and Computing, The University of Western Australia, Crawley, WA 6009, Australia; pejman.rowshanfarzad@uwa.edu.au (P.R.); martin.ebert@health.wa.gov.au (M.A.E.); 2Institute of Medical Physics, School of Physics, The University of Sydney, Camperdown, NSW 2050, Australia; annette.haworth@sydney.edu.au; 3Centre for Advanced Technologies in Cancer Research (CATCR), Perth, WA 6000, Australia; 4Department of Radiation Oncology, Sir Charles Gairdner Hospital, Nedlands, WA 6009, Australia; 55D Clinics, Claremont, WA 6010, Australia; 6School of Medicine and Population Health, University of Wisconsin, Madison WI 53706, USA

**Keywords:** focal boost radiotherapy, prostate cancer, intra-prostatic lesion, treatment planning, clinical trials

## Abstract

**Simple Summary:**

By delivering escalated doses to biologically defined sub-volumes within a target, focal boost radiotherapy is expected to enable an elevated tumor control without an increase in toxicity in prostate cancer. Evidence for the efficacy and variability of this approach has been summarized and evaluated to support future focal boost radiotherapy studies and trials. For 34 planning studies and 35 clinical trials, published during 2013–2023, the methodology of treatment planning, dosimetric outcomes, and clinical efficacy were summarized and evaluated in this review. The challenges reported in the reviewed studies and their potential solutions are highlighted to serve future focal boost radiotherapy studies and trials.

**Abstract:**

Background: Focal boost radiotherapy was developed to deliver elevated doses to functional sub-volumes within a target. Such a technique was hypothesized to improve treatment outcomes without increasing toxicity in prostate cancer treatment. Purpose: To summarize and evaluate the efficacy and variability of focal boost radiotherapy by reviewing focal boost planning studies and clinical trials that have been published in the last ten years. Methods: Published reports of focal boost radiotherapy, that specifically incorporate dose escalation to intra-prostatic lesions (IPLs), were reviewed and summarized. Correlations between acute/late ≥G2 genitourinary (GU) or gastrointestinal (GI) toxicity and clinical factors were determined by a meta-analysis. Results: By reviewing and summarizing 34 planning studies and 35 trials, a significant dose escalation to the GTV and thus higher tumor control of focal boost radiotherapy were reported consistently by all reviewed studies. Reviewed trials reported a not significant difference in toxicity between focal boost and conventional radiotherapy. Acute ≥G2 GU and late ≥G2 GI toxicities were reported the most and least prevalent, respectively, and a negative correlation was found between the rate of toxicity and proportion of low-risk or intermediate-risk patients in the cohort. Conclusion: Focal boost prostate cancer radiotherapy has the potential to be a new standard of care.

## 1. Introduction

Prostate cancer (PCa) is the second most common cancer worldwide in males. Approximately 2.3 million new cases and 740,000 deaths are expected globally by 2040 [1,2]. Radiotherapy has firmly established its position as the primary modality for the treatment of prostate cancer. However, conventional PCa radiotherapy, which involves prescribing homogeneous dose distributions, does not account for the multifocality and the inter- and intra-tumor heterogeneity that encompass a wide range of biological characteristics [3,4,5]. Previous studies have indicated that local failure within the prostate following conventional radiotherapy commonly originates from the original tumor site [6,7]. As a result, it has been suggested that the patient’s biological information be incorporated into treatment planning to judiciously target higher doses to high risk regions and enhance tumor control [8]. The concept of the biological target volume (BTV) was first introduced by Ling et al. [9] which refers to functional sub-volumes within a target. A focal boost technique involves prescribing an escalated dose to the BTV, whilst the dose prescribed to the remaining target volume remains unchanged or de-escalated [10,11,12,13]. In the case of prostate cancer, the BTV is typically referred to as the intra-prostatic lesion (IPL), or intra-prostatic lesions (IPLs) in the instance of multi-focal disease. The volume receiving the escalated dose may include an additional margin around the IPL to account for uncertainties in delineating the IPL, and this expanded volume may be referred to as the “boost volume” or “focal volume”. Building on the reported benefits of whole-gland dose escalation on tumor control [14,15,16], it was hypothesized that the focal boost technique can improve treatment outcomes without increasing toxicity [9,17,18,19,20].

Studies included in this review article included those that incorporated a dose escalation to the IPL or boost volume, with the aim being to summarize the evidence for the efficacy and variability of this approach. For studies included in this review, the following were examined: the methodology of identification of IPLs, treatment planning techniques, reported dosimetric results, dose-limiting factors, comparison of IPL focal boost across different treatment modalities, reported clinical outcomes from focal boost trials, and current challenges associated with focal boost radiotherapy.

## 2. Methods

### Study Inclusion Criteria

To search and synthesize the methodologies and results reported from both focal boost planning studies and trials, a literature search was conducted using OneSearch, with the most recent search performed on 3 March 2023. The search terms included: “intraprostatic” AND “radiotherapy” AND title contains “prostate”. The selected resource type was “Articles”, and the language was limited to “English”. To ensure up-to-date information and align with current advancements in treatment equipment, radiobiology, medical imaging, and reporting standards for clinical results, studies published prior to 2013 were excluded from the review. For consistency throughout this review, the gross tumor volume (GTV) is defined as the volume of IPLs, and the boosted planning target volume (PTV_boost_) is defined as the expansion from GTV that encompasses IPLs, potential microscopic disease, and GTV motion.

In addition to the criteria mentioned above, the identified studies were required to meet the following conditions: The studies had to include the defined GTV within the prostate gland.The studies had to incorporate dose escalation specifically to the GTV, in comparison to the prescribed dose administered to the remaining prostate.

The planning studies and trials were differentiated based on whether each study reported patient follow-up. For planning studies, the surveyed characteristics included the number of patients, modality used for identifying the GTV, method employed for defining target volumes, used techniques for GTV determination, dose prescription to the target volume, treatment modality(ies) utilized for GTV dose escalation, and the planning results reported through dose metrics. In addition to the characteristics recorded for planning studies, the composition of the investigated cohort was surveyed in terms of risk groups, use of hormonal therapy, initial median prostate-specific antigen (PSA) levels, median follow-up time, acute/late grade 2 or greater (≥G2) genitourinary/gastrointestinal (GU/GI) toxicity, as well as clinical and dosimetry outcomes.

Meta-analyses were conducted across planning studies to determine the correlation between the volume of GTV and reported dose metrics. Furthermore, correlations between acute/late ≥G2 GI or GU toxicity and factors such as risk group, initial median PSA level, and the proportion of cohort treated with hormone therapy were computed across the results of trial studies. The strength of these correlations was quantified using Spearman’s correlation coefficient.

## 3. Topical Review

Figure 1 provides a flowchart for the selection of reviewed studies.

### Statistics of Reviewed Studies

After assessment for eligibility, a total of 34 planning and 35 clinical trial publications were included from a total of 462 records identified through OneSearch. It should be noted that two publications reported the clinical results of the same trial [21,22]. Additionally, two other publications [23,24] reported the same trial but with a different number of patients in the cohort. Therefore, these two publications are considered to represent two separate trials, bringing the total number of trials included to 34. The summarized characteristics of the reviewed studies are presented in Appendix A.

As shown in Table 1, the focal boost was predominantly administered to intermediate- and high-risk patients in all 34 reviewed trial studies. For 32 cohorts totaling 2000 study participants across 23 trials that reported median PSA, the average reported median PSA was 9.08 ng/mL. The highest and lowest reported median PSA were 16.3 [25] and 5.1 [26], respectively. In the majority of trials, IPLs were delineated using multi-parametric MRI (mpMRI) to define the GTV [21,22,23,24,25,26,27,28,29,30,31,32,33,34,35,36,37,38,39,40,41,42,43,44,45,46,47,48], while the GTV was identified by positron emission tomography (PET-CT) in 2 trials [49,50] and two trials employed both mpMRI and PET-CT [51,52]. Three trials utilized transrectal ultrasound (TRUS) to identify the GTV [53,54,55]. Dose escalation to the GTV was achieved using external beam radiotherapy (EBRT) in most trials (which includes intensity modulated radiotherapy (IMRT), volumetric modulated arc therapy (VMAT), and CyberKnife) (Appendix A). In seven trials, the GTV boost was achieved with high-dose-rate (HDR) brachytherapy [34,37,40,45,47,54,56]. Additionally, three trials utilized low-dose-rate (LDR) brachytherapy for GTV dose escalation [26,53,55]. Two studies conducted by Zamboglou et al. [52] and Sanmamed et al. [34], employed both EBRT and HDR brachytherapy for GTV dose escalation. Regarding the median follow-up time, a total of 41 cohorts from the 34 reviewed trials reported this information. The median reported median follow-up time across these cohorts was 39 months. The longest reported follow-up time was 124 months [49].

In the planning studies, as shown in Table 2, the majority of studies utilized mpMRI for GTV delineation [10,18,45,57,58,59,60,61,62,63,64,65,66,67,68,69,70,71,72,73,74,75]. Five studies utilized PET-CT [12,76,77,78,79,80], and four studies used “other methods” such as histopathology data or hypothetical IPLs [81,82,83,84]. Regarding the GTV identification in the study by Zamboglou et al. [83] it involved the usage of mpMRI, and PET-CT, and histopathology data. The considered treatment modalities across the planning studies included IMRT, VMAT, intensity modulated proton therapy (IMPT), CyberKnife, and helical Tomotherapy (Tomotherapy) in 17, 11, 3, 2, 7, and 1 study, respectively (Appendix A).

## 4. Overview of Planning Methodology

### 4.1. GTV Identification

#### 4.1.1. GTV Identification by MRI

In the majority of studies (51 out of 68), mpMRI was utilized for manual delineation of GTV. There is ongoing debate regarding the optimal method for GTV determination in mpMRI [58]. However, it has been recommended to use multimodality imaging for this purpose [82]. The combination of T2 or T1 weighted MRI (T2w/T1w), along with the diffusion-weighted MRI (DWI) has shown to be highly sensitive in identifying IPLs [85,86]. Given the complementary nature of apparent diffusion coefficient (ADC) and volume transfer constant (K_trans_) distributions (i.e., neither ADC nor K_trans_ distributions alone can fully provide the necessary information for GTV identification) [86], the combination of DWI and dynamic contrast-enhanced MR perfusion (DCE) has been reported to achieve higher sensitivity and specificity from 70% to 87% [87,88]. Adding DCE to DWI and T2w imaging has been shown to improve tumor detection accuracy at the voxel level compared with DWI and T2w alone [89]. As a result, reviewed studies commonly utilize both DWI and DCE within imaging protocols to enhance the accuracy of GTV delineation [85,90] by contouring the volume based on low ADC and high K_trans_ signals [61]. Moreover, the accuracy of GTV delineation using mpMRI can be even further improved by incorporating other supportive information, such as the distribution of the Gleason scores described on pathology [72] or the position of IPL indicated by the biopsy [77,78].

For the brachytherapy focal boost studies, MR spectroscopic imaging (MRSI) was utilized to map the distribution of voxel-wise cellular activity for GTV delineation [68,91,92]. MRSI data were co-registered with MRI images and segmented into multiple voxels based on the resolution of the MRSI data. GTVs were then determined by assigning a grade to each voxel on a scale of 1 (benign) to 5 (malignant) based on the presence of metabolite markers [91]. It was also suggested that combining MRSI with DCE can further enhance the detectability of GTVs [92]. The detectability of GTV is optimized when using the combination of all three imaging modalities (DWI, DCE, and MRSI) [93].

#### 4.1.2. GTV Identification by PET-CT

Seven studies (7/68) utilized PET-CT to define the boost volume. In these studies, the GTVs were created by applying a threshold to the voxel-wise standardized uptake value (SUV) distribution, either using the maximum SUV (SUV_max_) or the ratio to the SUV of the background within the region of interest (ROI). 

The proportion of SUV_max_ used as a threshold varied across the studies. Zamboglou et al. [52,83] used thresholds of 20% and 30% of SUV_max_ [94,95], whilst Thomas et al. utilized a threshold of 40% [76]. Based on the previous reported tumor-to-background ratio (TBR) for ^18^F-choline tissue uptake [96,97], Kuang et al. [78] defined the threshold as 60% or 70% of SUV_max_. Chang et al. [79], in a comparison of delineated IPL contours from histology data with five other methods (visual, PET Edge, Region Grow, absolute SUV thresholds, and percentage of maximum SUV thresholds), found the threshold of 60% SUV_max_ showed the best correlation with the histology-defined IPL contour, although its advantage was not statistically significant [80]. 

Pinkawa et al. [11,77] defined the GTV by applying a threshold to PET-CT data using a choline TBR > 2 [96,98,99]. The background SUV was determined by the maximum SUV in an area that with the lowest activity within the prostate [11]. Kwee et al. [97] suggested that the PET scan should be conducted 1 h after injection, and the optimal mean TBR was reported to be 1.8.

The SUV threshold should be selected appropriately in line with the treatment goal, as a higher threshold would improve the specificity of the defined GTV and omit smaller tumor areas [11]. For example, it was reported that a threshold of 50% of SUV_max_ is sufficient to cover all biopsy-positive sextants [78]. Conversely, a lower threshold would define a larger volume for focal boosts making it more difficult to meet the constraints of nearby organs at risk (OARs). 

Compared with the absolute value of the SUV threshold, a relative threshold was more commonly used for GTV identification, as the measured absolute SUV has limited repeatability due to changing parameters, such as competing transport effects and time of SUV evaluation [100]. Furthermore, owing to the decreased cell metabolism resulting from neoadjuvant treatments such as androgen deprivation therapy (ADT), the measured absolute SUV may be lower than in hormone naïve patients [101]. In contrast, the use of a relative threshold is less influenced by the SUV reduction in both tumor and background (non-tumor volume). Nevertheless, Pinkawa et al. found no significant differences in SUV_max_ between patients with and without neoadjuvant ADT in their investigation [77].

### 4.2. Comparison and Utilization of MRI and PET-CT in GTV Identification

For studies that reported the mean volume of the GTV and utilized PET-CT for GTV identification, the average reported volume was 6.19 cc (SD: 0.45 cc, range 5.4 cc [12] to 7.3 cc [83]). For the GTVs identified by mpMRI when the imaging sequences include DWI and/or DCE, the average GTV was 3.23 cc (SD: 1.22 cc, range 0.5 cc [59] to 6.2 cc [57]). GTVs defined by mpMRI were significantly smaller (Wilcoxon rank sum, *p* << 0.05) than those defined by PET-CT, in agreement with Zamboglou et al. [52,83]. 

GTV delineation using both PET and MRI have therefore been recommended and reported to have a high sensitivity [83,102,103]. By comparing GTVs defined by PET (GTV-PET), MRI (GTV-MRI), and the union of PET and MRI (GTV-union) with those defined by histology data (GTV-histo), Zamboglou et al. [83] found that GTV-union overlapped more with GTV-histo (percentage volume of GTV-histo overlapped: 93%) than GTV-PET (86%) and GTV-MRI (74%). Moreover, with the same focal boost prescription, the tumor control of GTV-union boosting was significantly higher than GTV-PET or GTV-MRI boosting alone with no or minimal increase in normal tissue complication probability (NTCP). In terms of acute toxicities and quality of life, boosting to both PET-CT and mpMRI defined GTVs was found to be feasible and safe [52]. It should also be noticed that because either of MRI and PET-CT can generate the volume required to fully cover the volume of the GTV defined from histopathology data [83], for studies that did not include multimodality imaging, the incomplete coverage of delineated IPL structures may lead to an overestimation of TCP or other treatment outcomes [104].

Because unifocal cancer occurs in 13–33% of patients [105], and 40% to 80% of IPLs in patients with multifocal tumors are clinically insignificant (volume less than 0.5 mL) [81], the number of boosted IPLs reported across the studies was mostly less than 3, with the majority reporting just one IPL. 

### 4.3. GTV Margin

Excluding brachytherapy focal boost planning (n = 7) and trial studies (n = 2), Table 3 summarizes the distribution of studies by GTV-PTV_boost_ margin size. Adding a margin to the GTV in treatment planning is not standard. Fifteen studies did not apply a margin to the GTV in focal escalation plans, and it was suggested that the intra-fraction motion of the GTV can be sufficiently covered by the dose fall-off of the boost volume, as a significant dose boost to 2–3 mm away from the GTV was reported [65,72]. It has also been suggested that intra-fraction uncertainties can be minimized by other techniques such as prostate tracking and compensation with a dynamic multi-leaf collimator [106].

For studies using a margin to account for inter- and intra-fraction motion, there was no agreed margin size for the GTV in focal boost studies. In most studies, a 5 mm margin (sometimes with 3 mm posteriorly) was used. This margin size was derived from previously published SBRT treatment protocols and results [86,107,108], where a 5 mm margin was shown to sufficiently cover subclinical extra-prostatic extension and treatment delivery uncertainties [72,85,109,110,111]. The largest margin applied was 10 mm in the research conducted by McDonald et al. [28], where the GTV was expanded with a 5 mm margin to form the CTV_boost_, and then the PTV_boost_ was formed by adding another 5 mm margin. Researchers who used cine-MRI [112], ultrasound [113], and fiducial seed matching [114,115] for prostate intra-fraction motion measurement recommended a smaller margin of 3 mm [63]. However, Maggio et al. [75] addressed concerns that a GTV margin of less than 5 mm may lead to a significant risk of missing the target, based on their experience using Tomotherapy [116,117]. Nevertheless, considering the dose fall-off from the boost volume, the applied margin could be smaller [65].

### 4.4. Dose Prescription

For studies using EBRT as the focal boost technique, the ratio between the biologically effective doses (BEDs, alpha-beta ratio = 3.1) of the reported GTV and the prescribed dose to the prostate is plotted in Figure 2. On average, the boost doses to GTVs were 135.9% and 126.3% of the prescribed doses to the remaining prostate for planning and trial studies, respectively. The highest prescription in planning studies was 273.5% [63] and the highest proportion in trials of 202.2% involved prescribing up to 50 Gy physical dose (BED: 211.2 Gy) to the GTV while 33.25 Gy (BED: 104.6 Gy) was prescribed to the rest of the prostate volume in 5 fractions [39].

To determine the correlation between GTV and dose coverage in focal boost plans, planning studies that reported the average volume of GTV and the metrics of D_95_ (dose received by 95% of the target volume) and/or D_mean_ (average dose level received by the target volume) of GTV and/or prostate were selected. By normalizing the reported dose metrics of D_95_ and D_mean_ with BED, the Spearman correlation coefficient revealed a very weak ($0<\left|correlation\right|<0.2$) and weak ($0.2<\left|correlation\right|<0.4$) negative correlation between the mean volume of GTV to D_mean_ (correlation = −0.19%) and D_95_ (correlation = −0.24%) of the prostate, respectively. However, the mean volume of GTV showed a moderate ($0.4<\left|correlation\right|<0.6$) and very strong ($0.8<\left|correlation\right|<1$) negative correlation with D_mean_ (correlation = −0.60%) and D_95_ (correlation = −0.89%) of the GTV, respectively.

### 4.5. Derivation of Dose Prescription from Conventional Radiotherapy

A number of dose escalation studies have determined improvement in tumor control; however, the optimal dose that results in improved tumor control without excess toxicity is important to determine, and it can be derived from dose prescriptions that are utilized in conventional uniform-dose radiotherapy. For instance, the RT01 trial [118] demonstrated an 11% improvement in biochemical relapse-free survival (bRFS) with a 15% increase in prescribed dose. Based on the hypothesis that a 15% escalation in GTV dose could lead to similar improvement in bRFS, the DELINEATE Trial [23,24] prescribed 82 Gy in 37 fractions to the GTV. The trial also included an arm that prescribed an iso-effective dose of 67 Gy in 20 fractions to the GTV.

The escalation of PTV_boost_ dose to 50 Gy (in 5 fractions) was determined by evaluating its feasibility and safety from a traditional 3 + 3 design that applied in the trials of Herrera et al. [30]. In this study, all patients were divided into three groups of three, and only when no dose-limiting toxicities (DLTs) were observed in the previous group, the higher boost level to PTV_boost_ was prescribed to the subsequent 3-patient group.

### 4.6. Derivation of Dose Prescription by Radio-Biological Features

Studies also employed radio-biological optimization to determine the boost dose of the GTV. In the study of Kazi et al. [67] the TCP calculations used the total numbers of clonogens in the entire prostate obtained from the datasets [119,120] and assumed a 9:1 ratio of clonogen in the GTV to the rest of the prostate [82]. To make the TCP of the conventional (whole gland) and the HDR focal boost treatments equivalent, a focal boost prescription of 7.5 Gy and 15 Gy to prostate and GTV, respectively, was derived for focal boost HDR brachytherapy to achieve an equivalent TCP of a conventional 10 Gy whole-gland treatment.

To determine the optimal dose to the GTV, Azzeroni et al. [121] calculated the probability of complication free tumor control, P+. P+ was incorporated into treatment planning as an objective function to find the optimal solution in the trade-off between TCP and NTCP. It was calculated as follows:$$P+=TCP\left(1-NTCP\right)\;Or\;P+=TCP-NTCP$$

Similarly, for the focal boost EBRT with 41 fractions, Seppälä et al. [12] introduced the probability of uncomplicated control (PUC) to determine the optimal boost dose to the GTV. PUC represents the probability of local tumor control without normal tissue complication, and expressed by the following equation [122]:PUC=TCP−P+δP(1−TCP)

The term “*P*” represents the probability of injury, while “δ” is equal to 0.2, representing the fraction of patients for whom local tumor control and radiation-induced injury are statistically independent [12,122]. By assuming a uniform clonogen cell density distribution for the IPLs and utilizing PUC as an objective function, Seppälä et al. [12] reported that the TCP of PTV_boost_ reached a plateau when a prescription dose of 84 Gy was administered to the PTV_boost_, and the highest PUC of PTV_boost_ was achieved with an average dose of 82.1 Gy to the PTV_boost_.

By defining only the maximum dose of 86 Gy (in 37 fractions) to the GTV, Uzan et al. [48] determined the optimal prescribed dose using inverse optimization with a series of predefined objectives. These objectives included maximizing the TCP of the GTV while ensuring that the NTCP limits from the conventional treatment plans without a focal boost were not exceeded. The inverse optimization process indicated that a boost dose of up to 100 Gy can be prescribed without causing additional toxicity. Similar methods were also employed by Onjukka et al. [27], where the treatment plans were optimized by maximizing the TCP without surpassing a median dose of 68 Gy to the PTV_boost_ or an NTCP of 5% for rectal bleeding and faecal incontinence.

A machine learning-like method was proposed in the research conducted by Rezaeijo et al. [36]. In this study, the ADC and K_trans_ data of the IPLs from 120 patients were utilized to train a hierarchical clustering model. Subsequently, the IPLs of 20 patients were categorized into three risk groups using the hierarchical clustering model, and a total dose of 80, 85, and 91 Gy was prescribed to low-, intermediate-, and high-risk IPLs, respectively.

## 5. Overview of Results

### 5.1. Dosimetric Outcomes

As expected, all reviewed studies that compared conventional and focal boost radiotherapy consistently showed a significant dose escalation to the GTV [45,58,76,123]. However, in the context of a focal boost, it remains debatable whether it is possible to achieve both an escalated dose to the boost volume and improved sparing of OARs. 

For plans that involved higher doses prescribed to the boost volume and lower doses prescribed to the rest of the prostate volume, similar or not significantly different doses to OARs were generally reported [12,45,65,68,76,77,123]. For instance, compared with a treatment with the whole-gland dose of 40 Gy in five fractions (homogenous dose escalation to the entire gland), significantly improved protection of the bladder and rectum was reported with focal boost plans that de-escalated the dose to the prostate (35.2 Gy to prostate PTV with 40 Gy to PTV_boost_ in five fractions) [70].

Focal boost plans without dose de-escalation to the entire prostate typically resulted in higher doses to OARs [11]. However, in patients with favorable anatomy (i.e., IPLs that were not adjacent or overlapping with OARs) or those who utilized treatment modalities to enable the sculpture of steep dose gradients, even without dose de-escalation to the rest of the prostate volume, the focal boost plans could boost GTV without significantly increasing OAR doses and NTCP [75].

Studies have shown that certain cohorts of patients may benefit from focal boost techniques. Murray et al. [58] found that the same dose prescription could be achieved in focal boost plans with or without the inclusion of proximal seminal vesicles (proxSV), with no significant differences in GTV doses. This suggests that patients with intermediate-risk PCa and a higher risk of SV invasion can benefit from focal dose escalation. Compared with focal boost plans with prescribed doses for both the boost volume and the prostate, Thomas et al. [76] reported that focal therapy, where only the focal volume (i.e., the volume equivalent to boost volume) was prescribed with dose, achieved higher doses to the IPLs and better protection for OARs. However, this technique may increase the risk of local failure if undetected IPLs are located outside the boost volume. It has been proposed as an option for low-risk patients who are currently managed by active surveillance [76]. Moreover, patients who receive a higher escalated dose to the boost volume are likely benefit more from focal boost treatment because the higher boost level is more likely to be planned or delivered for patients with favorable anatomy [75].

The focal boost technique has the potential to be combined with other treatment techniques. Ciabatti et al. [59] conducted a comparison between focal boost VMAT plans with and without strict dose constraints applied to critical structures related to sexual function, such as the penile bulb (PB), corpora cavernosa (CC), internal pudendal arteries (IPAs), and neurovascular bundles (NVBs), in a 12-patient cohort. They used the same dose prescription for the prostate and boost volume and found that the sexual-sparing approach achieved significantly better sparing of sexual-function-related OARs without compromising dose coverage to the prostate and boost volume. Another study compared focal boost brachytherapy plans with up to two additional HDR needles within the boost volume to those with a standard needle arrangement. The study found that a higher boost level to the boost volume could be achieved, but this was accompanied by slightly higher doses to the rectum [61]. In a different approach, Amini et al. [81] defined the biopsy positive prostate lobe as the boost volume. This method led to lower doses delivered to the adjacent critical structures compared with the conventional prescription without boosting the affected lobe. Since a lower dose was delivered to the opposite NVB, the risk of erectile dysfunction (ED) was also expected to be lower after treatment.

Dose constraints for the PB were not considered in most of the reviewed studies, as the current literature cannot sufficiently support improved potency-preservation by sparing the PB. Additionally, introducing extra PB dose constraints would pose an oncological risk if it required reducing margins or lowering the dose to the target volumes [124]. Nevertheless, MRI-delineated plans have shown a significant reduction in dose to the PB and rectal wall, as MRI provides more accurate delineation of the prostate than CT [125,126]. Furthermore, a lower dose to the PB can be taken into account by contouring the prostate on MRI data for focal boost plans without applying specific dose constraints [62].

Urethral doses above 80 Gy have been linked to an increased risk of urethral strictures [127]. However, focal boost studies, such as the Focal Lesion Ablative Microboost in Prostate Cancer (FLAME) trial [25,128], did not incorporate dose constraints for the urethra. Considering the results of previously conducted focal boost trials [10,18,19,20], Housri et al. [63] suggested that acceptable acute genitourinary toxicity can be expected without the use of urethral dose constraints.

### 5.2. Dose-Limiting Factors

One of the leading reasons for the focal boost approach not being feasible was in instances of overlap between the boost volume and OARs. In the research undertaken by Blake et al. [60], the boost volume of 83% of patients overlapped with surrounding OARs. In 42% of patients, the intended boost dose of 86 Gy was limited due to the overlap between the boost volume and the urethra. Dankulchai et al. [64] also reported that satisfying urethral dose constraints was challenging when the boost volumes were in close proximity to the urethral volume. For brachytherapy patients who have recently undergone transurethral resection (TURP) and have a residual cavity, alternate dose constraints were suggested rather than the typically used GEC/ESTRO and ABS recommendations for focal boost treatment [129,130]. 

The ability to achieve an escalated dose to the boost volume may be limited when the boost volume is within 1.5 mm from the rectum wall [72,131]. Using the Pearson correlation coefficient, Murray et al. [58] found that the most significant factor affecting the ability to achieve the intended PTV_boost_ dose (D_95_) was the margin around the boost volume that overlapped with the rectum. However, the minimum distance from the boost volume to- or overlapped volume with- the bladder or urethra did not show a significant correlation with PTV_boost_ dose (D_50_ or D_95_). 

Similarly, Azzeroni et al. [121] reported that due to the overlap and smaller distance between the boost volume and the rectum, tumor control may decrease as a result of underdosing the boost volume to spare OARs. They also observed a correlation between the expected tumor control of focal boost plans with both the distance between the GTV and the rectum and their volume of overlap [121]. Moreover, the dose fall-off from the neighboring boost volume also led to the escalation in both rectal dose and NTCP in focal boost plans [76]. Maggio et al. [75] suggested that, with Tomotherapy, it is possible to escalate the median dose of PTV_boost_ up to 120 Gy if there is no overlap between the boost volume and the rectum. However, Housri et al. [63] found no correlation between the feasibility of focal boosting and GTV location, which may attributed to the generally large GTV–rectum distance (mean of 5.7 mm) of their investigated cohort.

For patients with larger rectal volumes, there was a trend towards failure in delivered focal boost plans, even without violating the dose constraints [63]. However, no significant relationship was found between the hottest dose to the rectum (D1cc) and the overlap between the prostate PTV and rectum [72]. As IPLs are most commonly found in the peripheral zone (PZ) [18,66,82,132], by the sufficient distance between the IPLs and the bladder, it has been reported that bladder dose constraints can be relatively easily satisfied without compromising the dose to the boost volume [60].

Although Kim et al. [71] suggested the number of IPLs is not significantly correlated with the achieved boost level, attempting to boost two or more IPLs with anatomically distinct positions within the prostate can significantly escalate the difficulty of planning and delivery of focal boost radiotherapy [133]. The hip-to-hip patient width may also be correlated with the possibility of achieving the prescribed boost dose, as it was found that escalated doses are easier to achieve within OAR dose constraints due to the greater depth of prostate (resulting from increased hip-to-hip width). However, this dose-limiting factor was mentioned in only one study [63]. Surprisingly, the volume of the GTV appears to have a minor impact on the likelihood of reaching the prescribed boost dose, as reported by Kim et al. [71], Housri et al. [63], and Murray et al. [58].

### 5.3. Treatment Modality Comparison

A comparison between IMRT plans with seven equally spaced 6 MV coplanar beams and IMPT plans with two parallel-opposing lateral beams showed that IMRT achieved a more conformal dose distribution to the boost volume [84]. However, IMPT, with its spread-out proton Bragg peak (SOBP) and the sculpted steeper proximal and distal dose fall-off, enabled greater dose homogeneity within the boost volume while significantly reducing the dose to the rectum and bladder. It also provided comparable dose to the prostate and SV, and better sparing for the femoral heads and PB. Similar findings were reported in a plan comparison study conducted by Cambria et al. [57]. In addition, for IMPT utilizing passive scattering (PS), it was suggested that conformalty (i.e., PTV coverage) can be further escalated by applying pencil beam scanning (PBS) [134]. When comparing IMPT, IMRT, and VMAT with the same dose prescription, all modalities were found to be adequate for focal boost treatment. However, IMPT (two fields or five fields) achieved a higher boost level to the GTV while providing the best sparing to the bladder, rectum, and urethra.

In terms of speed of delivery, conformality, and OARs protection, VMAT has generally been shown to be superior to IMRT in PCa radiotherapy [62,135]. In focal boost radiotherapy, Uzan et al. [48] reported that a two-arc VMAT technique can create a more conformal dose distribution to the target volume compare with an 11-field IMRT technique. Furthermore, by comparing focal boost IMRT plans with 3-, 5-, and 7-fields, and VMAT plans, improved rectal sparing with VMAT was reported by Ost et al. [62], and the average treatment time of VMAT (117s) was also found significantly shorter than IMRT with 3-(169s), 5-(231s), or 7-fields (289s). By comparing IMRT focal boost plans with different field arrangements whilst maintaining OAR constraints, Ost et al. [62] also reported the doses delivered to the boost volume and prostate were lower for 3-field IMRT than those for 5- and 7-fields. Consequently, this indicated that for focal boost radiotherapy planned by IMRT, a larger number of fields was recommended to enable creation of the conformal dose distribution to the target volume. The study by Ost et al. [62] also compared treatment plan delivery with 6 and 18 MV photon beams, and found no significant difference between the energy levels, which is consistent with the findings reported by Aoyama et al. [136].

In a study by Tree et al. [72], 15 patients were planned with focal boost using double-arc VMAT and CyberKnife robotic radiosurgery. Both techniques used the same dose prescription derived from a conventional CyberKnife treatment protocol. The CyberKnife plans, with an average of 215 beams and 59 nodes, achieved higher D_95_ values for the boost volume than the VMAT plans. However, the CyberKnife treatments required a significantly longer delivery time than VMAT (46 min vs. 5.9 min). Nevertheless, if the CyberKnife-like tumor tracking system cannot be deployed in VMAT delivery, a larger margin size will be needed to account for intra-fraction motion. This will likely result in dosimetric results that are not as good as CyberKnife [72].

HDR brachytherapy has the potential to be the superior technique for achieving a focal boost due to its dosimetric properties, which allows for highly conformal dose distributions for GTVs close to dose-limiting OARs [34]. However, this advantage must be balanced with the inconvenience and labor requirements associated with HDR brachytherapy, such as hospitalization, anesthesia, and invasive surgery [32].

### 5.4. Toxicity

For the reviewed trials, acute ≥G2 GU and GI, and late ≥G2 GU and GI toxicities were reported for 20, 23, 22, and 23 cohorts, respectively. Table 4 lists the total number of patients in the cohorts for which the corresponding toxicity was reported, as well as the number of patients experiencing the toxicity. Acute ≥G2 GU toxicity was the most prevalent, while the incidence of late ≥G2 GI toxicity was reported the least. Aluwini et al. [32] found the most common acute GU toxicities were urinary urge and increased night voiding frequency, and for GI toxicity, it was increased stool frequency. Overall, ≥G2 GU toxicity was more common than ≥G2 GI toxicity [23,35].

The Spearman correlation between acute/late ≥G2 GU or GI toxicity and the proportion of patients in each risk group, the initial median PSA, or the proportion of patients who received hormonal therapy is listed in Table 5. From the table, a negative correlation was found between the rate of toxicity and proportion of low-risk or intermediate-risk patients in the cohort. The correlations between the proportion of low-risk patients and acute ≥G2 GU and late ≥G2 GI were found to be moderate ($0.4<\left|correlation\right|<0.6$). The proportion of high-risk patients had a positive but weak correlation with toxicity, indicating that an increased proportion of high-risk patients in the treatment cohort might result in higher toxicity. The correlations between toxicity and the proportion of patients receiving hormonal therapy or the initial median PSA were very low.

### 5.5. Clinical Efficacy 

#### 5.5.1. Conventional Fractionation Focal Boost Trials

Compared with conventional radiotherapy, focal boost radiotherapy has been reported to improve treatment outcome without significantly increasing toxicity [25,32]. The FLAME trial [25] is the randomized controlled trial (RCT) with the largest number of patients conducted so far. In this trial, 287 patients were assigned to the standard arm (receiving 77 Gy to prostate PTV in 35 fractions) while 284 patients were assigned to the focal boost arm (receiving 77 Gy and 95 Gy to prostate PTV and GTV respectively in 35 fractions). The results showed a significant improvement in 5-year biochemical disease-free survival (bDFS) in the focal boost arm, with similar prostate cancer-specific survival and overall survival (OS) between the two arms. Kaplan–Meier analysis and Cox regression indicated that the focal boost arm was also expected to have significantly higher 7-year bDFS and disease-free survival (DFS) with half the rate of biochemical failures (BF) compared with the standard arm [25]. Furthermore, in comparison with the results reported in trials of whole-gland dose escalation [118,137,138], the focal boost arm of the FLAME trial demonstrated a higher 5-year bDFS. For instance, in the Androgen Suppression Combined with Elective Nodal and Dose Escalated Radiation Therapy (ASCENDE-RT) trial [139], whole-gland boost using LDR brachytherapy (^125^I) resulted in increased estimated 5-, 7-, and 9-year bDFS according to Kaplan–Meier analysis. However, it was associated with higher toxicities compared with whole-gland boost by EBRT. The FLAME trial, on the other hand, showed improved clinical outcomes and similar toxicity rates to the standard arm, suggesting that focal boost radiotherapy can achieve comparable treatment outcome to whole-gland LDR brachytherapy boost in ASCENDE-RT without the added toxicities [25]. Logistic regression analysis indicates that the incidence of distant metastatic failure and BF up to 7 years decreased with increasing boost level to the GTV in the focal boost arm, although the benefit of GTV dose escalation may diminish around 95 Gy [25]. Additionally, in comparison with other hypo-fractionation trials such as the CHHiP (conventional versus hypofractionated high-dose intensity-modulated radiotherapy for prostate cancer) trial [140] and the HYPRO (hypofractionated versus conventionally fractionated radiotherapy for patients with prostate cancer) trial [141], the FLAME trial reported similar rates of toxicity [142]. These findings support the safe implementation of focal boost radiotherapy in routine clinical practice for patients with localized intermediate- and high-risk prostate cancer [142].

A cohort of 60 patients, mostly intermediate-risk (n = 30), was evaluated by Zapatero et al. [35] using post-treatment mpMRI to assess treatment response. In 83% of patients, the treated IPLs had disappeared on both DWI and T2w images 6 months after administering an 80 Gy focal boost to PTV_boost_ (with 76 Gy to the prostate PTV) in 35 fractions. The remaining 17% of patients had reduced volumes of treated IPLs, which were undetectable on mpMRI imaging by 9 months after treatment. Moreover, the focal boost treatment showed a more favorable toxicity profile than whole-gland boost treatment (80 Gy to prostate PTV without boost to GTV [143]) [35].

Additionally, post-treatment PSMA-PET enables the detection of biochemical recurrence with high accuracy [144]. A meta-analysis of 29 trials [145] found that [^68^Ga]-PSMA-11 has high specificity and sensitivity of 0.96 (95% CI: 0.85–0.99) and 0.74 (95% CI: 0.51–0.89), respectively, for the assessment of nodal disease. In men with biochemical recurrence, the positive predictive value of PSMA-PET is also high, at 99% [145].

In the trial conducted by Kuisma et al. [49], escalated median doses of 80.4 Gy were prescribed to PET-CT defined GTVs in 38 fractions (76.6 Gy to the prostate). The study compared the treatment outcomes of 19 patients who remained recurrence-free, and 11 patients who experienced recurrences (either biochemical, local, and/or distant failure). Surprisingly, there were no significant differences in the delivered doses between the two groups of patients, indicating that the GTVs of the recurrence group were not underdosed. In terms of treatment results, Kuisma et al. found that the volume of IPLs mattered more than the number of IPLs, and the number of IPLs did not have a statistically significant impact on progression-free survival (PFS) or OS. One hypothesis is that radioresistant hypoxic cells are present in higher numbers in IPLs with larger volumes [146]. Additionally, the study reported a statistically significant difference in OS and PFS between different risk groups, with patients with higher-risk disease having lower survival times.

#### 5.5.2. Extreme Hypo-Fractionation Focal Boost Trials

Alayed et al. [41] compared treatments with and without focal escalation to the GTV with extreme hypo-fractionated radiotherapy that included elective pelvic nodal irradiation. The control group received 35 Gy to the prostate and 25 Gy to the pelvis in five fractions, while the treatment group received an additional 50 Gy boost to the GTV. The study found no significant differences in cumulative late GI or GU toxicity between the two groups. However, the control group demonstrated significantly higher acute G2 and lower acute G1 GI toxicity, although there were no significant differences in the proportions of patients experiencing acute ≥G2 GI or GU toxicity between the two groups.

In the study by Herrera et al. [30], 20 patients were prescribed 36.25 Gy to the prostate PTV and up to 50 Gy to the PTV_boost_ in five fractions. The study reported that only 25% of patients experienced acute G1 or G2 GI toxicity lasting more than 90 days. This was lower than the previously reported acute GU toxicity rates in trials where the entire prostate was prescribed with 40–50 Gy in five fractions. The authors suggested that the lower rate of toxicity may be due to better rectum sparing achieved through the use of a mandatory rectal spacer, as also reported in Alayed et al. [41].

Among the reviewed studies with extreme hypo-fractionation schedules, the trial conducted by Hannan et al. [31] prescribed the highest boost dose. In this trial, 50–55 Gy, 47.5 Gy, and 22.5–25 Gy were prescribed to PTV_boost_, PTV of the prostate (+ ProxSV), and PTV of the pelvic lymph nodes in five fractions, respectively. The trial reported clinical outcomes, including a 2-year actuarial biochemical control rate of 96.6%, biochemical progression-free survival of 94.8%, disease-specific survival of 100%, and OS of 98.2%. The inclusion of elective pelvic lymph nodes in dose coverage for high-risk patients remains controversial. Some trials reported improved biochemical failure-free survival and distant metastasis-free survival with pelvic lymph nodes inclusion [147], while no treatment benefit was found by other studies [148,149]. Including pelvic lymph nodes may result in higher GI toxicity [31,149]. However, the trial by Hannan et al. [31] reported a favorable toxicity profile, and the rates of acute G2 GU and GI toxicity at 90-day were comparable to other extreme hypo-fractionation trials [150,151]. The authors attributed these outcomes to the systematic use of rectal spacers, prophylactic steroids (and blockers), evidence-derived constraints, stereotactic setup, small PTV margins, and effective image guidance.

In a cohort of 64 patients who received a 64.5 Gy boost to the PTV_boost_ in five fractions, Marvaso et al. [33] observed that patients with a bladder volume lower than the median of the cohort (341 cc) experienced a worsening of urinary symptoms after treatment, and these symptoms were not resolved during the entire follow-up period. In contrast, patients with a bladder volume higher than the median showed a trend towards a decrease in the progression of urinary symptoms with a median recovery time of 12 months. However, no correlation was found between bladder volume and ≥G1 GU toxicity. In contrast, the study found an association between ≥G1 GI toxicity and the patient’s rectum volume [33].

In the phase II hypo-FLAME trial [39], the GTVs were prescribed 35 Gy, with an iso-toxic boost up to 50 Gy in five fractions. The trial successfully delivered a median mean dose of 44.7 Gy (range: 37.7–50.9 Gy) to GTVs, with 99% of the GTVs receiving a median dose of 40.3 Gy (range: 36.2–50.7 Gy) [39]. Compared with the toxicity of reported conventionally fractionated radiotherapy without focal boost [152], the hypo-FLAME trial showed a considerably lower ≥G3 GU and GI toxicity. Additionally, it reported lower ≥G2 GU and GI toxicity compared with the FLAME trial [142]. The improved toxicity profile was attributed to the prioritization of OAR dose constraints over the boost level to the GTV.

#### 5.5.3. Brachytherapy Focal Boost Trials

In contrast to the positive results observed with multi-fraction focal boost treatment, single fraction HDR brachytherapy with a 21 Gy focal boost to the GTV did not improve treatment outcome compared with whole-gland 19 Gy single fraction HDR brachytherapy [37]. This disappointing outcome may be due to the hypoxia of tumor cells within the GTVs, as more than one fraction is required for tumor re-oxygenation to overcome hypoxia mediated radio-resistance. Additionally, a proportion of cells being in a resistant phase during that single fraction [37]. The limited ability of single-fraction HDR monotherapy to adequately escalate local control by using focal boost technique has also been highlighted by Alayed et al. [40]. Even for HDR brachytherapy without a focal boost, when comparing a single 19 Gy fraction to an expected biologically equivalent 27 Gy in two fractions, the single fraction group exhibited a significantly lower 5-year biochemical control rate [153].

In the trial carried out by Guimond et al. [53], LDR brachytherapy with and without a focal boost was compared. The cohort with a focal boost had a higher incidence of dyslipidemia and received significantly higher doses to the bladder and rectum. The focal boost cohort demonstrated a slightly higher estimated 7-year bDFS rate (96% vs. 89%). Furthermore, there were no statistically significant differences in toxicities between the two groups, which is consistent with the results reported by Gaudet et al. [154].

### 5.6. Challenges in Focal Boost Studies

Reduced treatment accuracy due to uncertainties in image registration has been reported when combining planning CT with mpMRI or PET-CT [18,58,60,66]. Rigid registration between CT and MRI images was reported to be inadequate for focal boost planning [60], and large image registration uncertainties have been observed in the superior and inferior directions [61]. To narrow the uncertainties in image registration, the iterative closest point method [155] was applied by Van Lin et al. [18]. When the fiducial markers were visible in both image modalities, the iterative closest point method registered images accurately by minimizing the root-mean-square distance between the surfaces of markers. 

For accurate image registration between MRSI and CT, mutual information-based automatic registration can be applied [156]. Deformable registration has been used to improve registration accuracy [27,57,58,60,157,158], as it accommodates variations in prostate size and shape. Nevertheless, deformable registration has several limitations and has not been verified in any focal boost trial [159].

For brachytherapy that utilizes MRSI and MRI for IPL detection, the registration error can be narrowed by undergoing medical imaging with the implanted catheters. This is because catheters provide additional markers for image registration, which can improve the registration accuracy. It has been recommended to use catheters together with fiducial markers to further improve image registration accuracy, particularly when fiducial markers are not easily identified on T2 MRI sequences [27,69]. However, catheters may impact the spectroscopic responses of the MRSI, owing to the trauma caused by catheter implantation [69].

If hormonal therapy was included in patient treatment, conducting both mpMRI imaging and GTV identification before hormone therapy is more appropriate [123], as the hormone therapy may make IPLs less conspicuous on mpMRI [60,61,160,161] by decreasing the contrast between IPLs and healthy prostate tissues [160,161,162,163]. 

The limited spatial resolution of PET can obscure small IPLs due to a partial-volume effect (PVE) [164]. Seppälä et al. [12] reported that IPL determination became problematic for identifying IPLs with diameters smaller than 2 cm due to the PVE, resulting in IPL contours that were larger than their actual volume.

Delivery of a spatially varying dose prescription in focal boost plans demands more reproducible and accurate prostate positioning than conventional radiotherapy. Image-guided radiation therapy (IGRT) with implanted fiducial markers [165] or a radio-opaque urethral catheter [166] was commonly proposed by studies to accurately reproduce prostate position during treatment [81,82]. In addition, immobilizers such as endorectal balloons (ERBs) [167,168] can also be used, but it was reported that due to the presence of surrounding gas and stool, the inter-fraction motion of the prostate can be great with the use of ERBs. Therefore, the use of fiducial markers and appropriate correction protocols is recommended when employing ERBs [169,170,171,172].

To shape the sophisticated dose distribution of focal boost plans, longer treatment times are expected in treatment delivery. Aluwini et al. [32] reported a slightly higher but not significant average treatment time for patients treated with a focal boost on Cyberknife (64 min) compared with those without a boost (59 min). However, if the appropriate tumor tracking system is not deployed, the potential problem of prostate intrafraction motion may arise due to the extended treatment time, particularly with treatment times exceeding 8 min [173,174]. Moreover, the use of immobilizers during treatment also increases the treatment time. For instance, endorectal balloons are estimated to extend the treatment time by 3 min per fraction [167].

The most frequently mentioned limitations in the reviewed trials are insufficient follow-up time [25,30,31,33,34,35] and limited participant numbers [30,33,34,35,36,52]. Other limitations include missing quality of life questionnaires due to emergency events (e.g., the COVID-19 outbreak) [52], an inhomogeneous distribution of patient characteristics between control and treatment groups [52], a lack of post-treatment histological confirmation, and lack of a comparison cohort [35]. 

### 5.7. Future Opportunities

The linear-quadratic TCP model has been used in studies for plan optimization and evaluation, but the assumption of homogeneously distributed tumor cell density within the GTV or prostate was generally made [12,67,121]. A pipeline has been proposed for deriving patient-specific cell density distributions from histology data [175,176], which has been integrated within a TCP estimation framework to guide a patient-specific optimal dose distribution [8,177]. To predict the cell density distribution for future patients, Finnegan et al. [178] generated a population-based statistical model by summarizing the cell density and tumor probability distribution of a cohort of 63 patients. Alternatively, radiomics can be used to build correlations between radiomics features and biological characteristics of the tumor or the biochemical and pathologic response. This enables the prediction of the distribution of biological information (e.g., tumor cell density, Gleason Score, or hypoxia) within the prostate or the probability of response [35,179,180,181,182,183,184]. The predicted biological information can be incorporated into plan optimization for focal boost radiotherapy, and with the innovative MRI-guided radiation therapy, further improvements in toxicity potentially can be achieved for focal boost treatment [31,185,186,187].

Post-treatment quantitative medical imaging or biopsy was recommended to be applied in trials for treatment response evaluation or radiotherapy efficacy prediction [35]. Even though very few studies included imaging or histopathology examination after the treatment, it was found that the biopsy-proven local control had a significant correlation with metastasis-free survival [188].

Several studies included in this review reported dose constraints and evaluation metrics for dose homogeneity (e.g., homogeneity or inhomogeneity coefficient/index [57,84] or target dose homogeneity [62]) for treatment planning and evaluation, respectively. There is debate regarding whether homogeneity of the dose distribution in the treated volume (e.g., prostate and GTVs) is beneficial for tumor control [189]. It has been suggested that a heterogeneous dose distribution may lead to a better treatment outcome, as heterogeneity in the dose distribution does not necessarily require sacrificing OAR sparing [190]. Furthermore, to allow gradients for GTV boosting and to maximize prostate doses, it has been suggested to remove limits on dose heterogeneity [30].

Although the IPL location may have a significant impact on rectal or urethral dose (with hence impact on GU or GI toxicity), few studies report the location of the IPLs [18,34,82,132,191]. The location of IPLs within the prostate can be approximately described in words (e.g., left-lateral-PZ) [18,82] or precisely described using the indexed lesion sites indicated on the prostate sector map by Weinreb JC et al. [192] or the modified sector map [132,191]. The proportion of IPL locations in the cohort can be plotted and presented in a figure [34].

Studies that did not include DWI or DCE for GTV delineation were excluded from the volume analysis of this literature review, as these sequences are preferred due to their higher accuracy for GTV identification. It is acknowledged, however, that differences in radiological techniques used to identify the size and location of the IPLs may lead to different conclusions, and the application of standardized imaging protocols and reporting of these protocols is required to avoid the issue of bias when comparing studies. In the toxicity analysis, due to the variations in follow-up time, toxicity grading, and reporting among the reviewed trials, all reported toxicities were assumed to be equivalent. However, the calculated toxicity correlation factors demonstrated good consistency with the clinical results reported in the trials. The correlations between mean or median volume of GTV and toxicities were not investigated because of the insufficient number of reported mean or median volumes for the GTV.

## 6. Conclusions

Current planning and trial focal boost studies have been reviewed in terms of planning methodology, dosimetric results, and treatment outcome. In reviewed focal boost studies, MRI and PET-CT were commonly used for GTV identification. The combination of multiple MRI sequences or that of MRI and PET-CT is recommended to achieve higher sensitivity and specificity in GTV delineation. Most studies did not add a margin to the GTV during treatment planning. However, for studies using a margin to account for inter- and intra-fraction motion, a 5 mm margin was mostly applied. There is no agreed dose prescription in focal boost therapy. Studies derived dose prescriptions either based on the prescriptions used in conventional radiotherapy or radio-biological models.

All reviewed studies that included a plan comparison between conventional and focal boost radiotherapy reported a significant dose escalation to the GTV. Especially for patients whose anatomy is favorable for focal boost technique, a better treatment outcome is expected. However, the feasibility of the focal boost approach may be compromised due to substantial overlap between the boost volume and OARs, the close distance between the boost volume and the rectum wall, a large rectal volume, and attempting to boost two or more IPLs with anatomically distinct positions within the prostate.

Focal boost radiotherapy has been reported to improve treatment outcome without significantly increasing toxicity compared with conventional radiotherapy. Acute ≥G2 GU toxicity and late ≥G2 GI toxicity were reported as the most and the least prevalent, respectively. Nevertheless, likely due to cell cycle dynamics and hypoxia, the single-fraction HDR monotherapy demonstrated a limited local control escalation by using the focal boost technique.

On the basis of positive results reported in the reviewed planning studies and trials, focal boost prostate cancer radiotherapy has the potential to be a new standard of care. With the continuing development of radiation oncology technologies and techniques, it is perceived that the case for focal boost radiotherapy will continue to grow. Such as when focal boost radiotherapy is incorporated with predicted biological information derived from machine learning models and a newly developed treatment technique, in MRI-guided radiation therapy high rates of tumor control are expected to be achieved without excess toxicity.

## Figures and Tables

**Figure 1 cancers-15-04888-f001:**
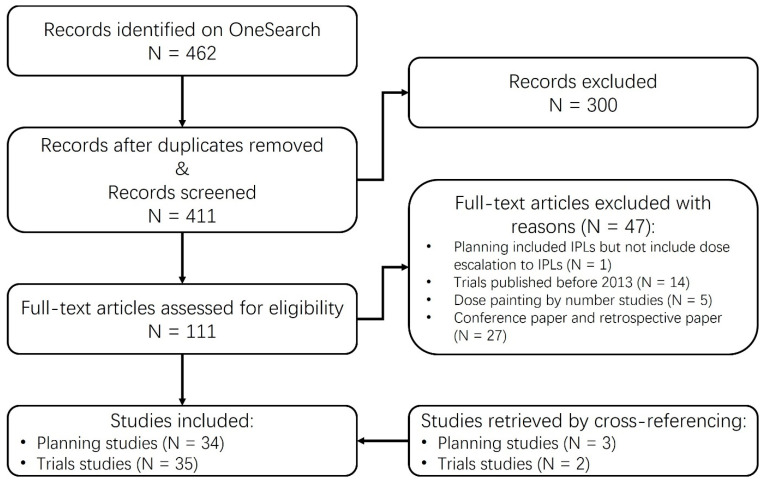
PRISMA flow diagram showing the study inclusion and exclusion process.

**Figure 2 cancers-15-04888-f002:**
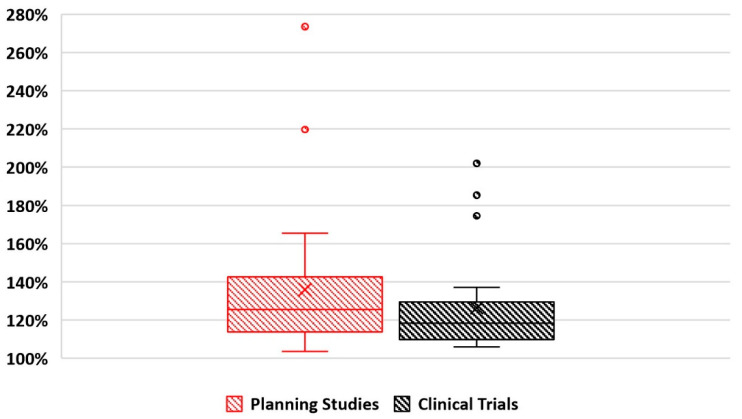
The ratio between reported GTV/boost volume and prostate/PTV prescribed BEDs for EBRT focal boost plans.

**Table 1 cancers-15-04888-t001:** Summary of characteristics reported by reviewed trial studies.

Number of Trial Studies Included	N = 34	
**Risk Group (patient number—proportion %)**		
Total included number of participants	n = 2919	100%
Low-risk	240	8.2%
Intermediate-risk	1196	41.0%
High-risk	1361	46.6%
Not reported	122	4.2%
**Median PSA**		
Average reported median PSA (ng/mL)		9.08
**Modality for GTV Identification (number of trials (%)—number of patients (%))**		
**N**	34 (100%)	2175 (100%)
**mpMRI**	27 (79.4%)	1953 (89.8%)
*DWI*	1	28
*T2w*	3	119
*T2w + DWI*	2	361
*T1w + T2w*	1	26
*T2w + DWI + DCE*	13	820
*T2w + T1w + MRSI*	1	47
*T1w + T2w + DWI + DCE*	1	25
*T2w + T1w +DWI + MRSI*	1	15
*T2w + DWI + DCE + MRSI*	1	225
*Sequence not reported*	3	125
**PET-CT**	2 (5.9%)	97 (5.0%)
**PET and mpMRI**	2 (5.9%)	162
**Others**	3 (8.8%)	125 (5.7%)
**Treatment modality**	**Number of Trials**	**Patients Treated/Boosted**
IMRT and/or VMAT	21	2277/1755
CyberKnife	3	78/78
CyberKnife + IMRT	1	25/25
HDR Brachytherapy	7	313/313
LDR Brachytherapy	3	226/116

**Table 2 cancers-15-04888-t002:** GTV identification modalities reported by reviewed planning studies.

Plan Studies Included.	34	*T2w + DCE + MRSI*	1
**MRI**	24	*T2w + DWI + DCE + MRSI*	1
*T1w + T2w*	1	*T1w + T2w + DWI + DCE + MRSI*	1
*T2w + DCE*	2	*MRSI*	1
*T2w + DWI*	4	**PET-CT**	5
*T2w + DCE + DWI*	6	*^68^Ga*	1
*T2w + T1w + DWI*	3	*^18^F*	2
*T2w + T1w + DWI + DCE*	1	*^11^C*	2
*T2w + MRSI*	3	**PET and mpMRI**	1
*T1w + T2w + MRSI*	1	**Others**	4

**Table 3 cancers-15-04888-t003:** Statistics of reported GTV-PTV_boost_ margin size for planning and trial studies.

	0 mm	2 mm	3 mm	4 mm	5 mm	6 mm	>6 mm
Planning	6	0	5	4	7	4	0
Trial	9	3	5	3	5	1	1

**Table 4 cancers-15-04888-t004:** Acute/late ≥G2 GU/GI toxicities reported by trials.

Toxicity		n	%
Acute ≥G2 GU	Total	1079	
	Positive	354	32.8%
Acute ≥G2 GI	Total	1409	
	Positive	203	14.4%
Late ≥G2 GU	Total	1196	
	Positive	231	19.3%
Late ≥G2 GI	Total	1421	
	Positive	148	10.5%

**Table 5 cancers-15-04888-t005:** The Spearman correlation between acute/late ≥G2 GU or GI toxicity and proportion of patients in each risk group, initial median PSA, or proportion of patients accepted hormone therapy.

	Acute ≥G2 GU	Acute ≥G2 GI	Late ≥G2 GU	Late ≥G2 GI
Low-risk %	−0.40	−0.16	−0.34	−0.43
Intermediate-risk %	−0.20	−0.20	−0.24	−0.16
High-risk %	0.25	0.24	0.31	0.24
Hormone therapy %	0.09	0.22	0.02	0.01
Median PSA	0.10	−0.01	0.17	0.14

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
