# Peer review of "Focal Boost in Prostate Cancer Radiotherapy: A Review of Planning Studies and Clinical Trials"

_cancers, 2023, doi:10.3390/cancers15194888_

Round 1

Reviewer 1 Report

The review entitled “Focal boost in prostate cancer radiotherapy: A review of planning studies and clinical trialse”, the authors summarize and evaluate the efficacy and variability of focal boost radiotherapy by reviewing focal boost planning studies and clinical trials that have been published in the last ten years. The manuscript well detailed and is convincing and would help advance the field of radiotherapy for prostate carcinoma by emphasizing focal boost prostate cancer radiotherapy has the potential to be a new standard of care. However, there are spelling errors throughout the manuscript. Therefore I advise consider revisiting manuscript by a native English language speaker.

There are spelling errors throughout the manuscript. Therefore I advise consider revisiting manuscript by a native English language speaker.

Author Response

Comments 1: The review entitled “Focal boost in prostate cancer radiotherapy: A review of planning studies and clinical trials”, the authors summarize and evaluate the efficacy and variability of focal boost radiotherapy by reviewing focal boost planning studies and clinical trials that have been published in the last ten years. The manuscript well detailed and is convincing and would help advance the field of radiotherapy for prostate carcinoma by emphasizing focal boost prostate cancer radiotherapy has the potential to be a new standard of care. However, there are spelling errors throughout the manuscript. Therefore I advise consider revisiting manuscript by a native English language speaker.

Response 1: Thank you for your comments, the manuscript has been revised by a native English speaker.

Response to Comments on the Quality of English Language
Comments 1: There are spelling errors throughout the manuscript. Therefore I advise consider revisiting manuscript by a native English language speaker.

Response 1: Thank you for your comments, the manuscript has been revised by a native English speaker.

Reviewer 2 Report

The authors did an interesting review of the prostate focal boost. They analyzed many articles, and they cover many different aspects: contouring, dose, technique, and toxicities.

I found it well done and useful.

I found it a little bit longer:  Is it possible to put together 2 studies, that are similar? Especially in the Clinical Efficacy chapter.

I suggest emphasizing also in the Conclusion the negative results of the Boost with Brachytherapy

Major issue:

  1. Pag4, line 150. The author wrote Robotic radiotherapy but in Table 1 they wrote only Cyberknife. The 2 voices have to be the same
  2. Pag 6 Line 209. The wrote in letter eight but in the bracket they wrote 7. Which is the correct one?
  3. Pag 8, Dose Prescription. They reported the dose only in percentage. I would like to total dose (first serie + focal Boost), that the Prostate received. The same issue fir the figure 2.
  4. Table 5: we do not need the % of the total, if there is always 100 %. They can format better the table

Reviewer 3 Report

Focal boost in prostate cancer radiotherapy: A review of planning studies and clinical trialse

Yutong Zhao 1,*, Annette Haworth 2, Pejman Rowshanfarzad 1 and Martin A. Ebert 1,3,4,5 See comments for authors:

The study discussed various methodologies for planning and prescribing doses in cancer treatment, with a focus on focal boost studies. It emphasized the importance of accurately defining the Gross Tumor Volume (GTV) using imaging modalities like MRI, PET-CT, and MR spectroscopic imaging (MRSI). The text also touched upon different margin sizes for GTV to account for motion and uncertainty. Furthermore, the study highlighted the significance of dose prescription, considering both conventional radiotherapy principles and radiobiological features. It provided a comprehensive overview of the methods employed in planning and prescribing doses for cancer treatment, stressing the need for precision and individualized approaches. In the context of comparing conventional and focal boost radiotherapy, the study found consistent evidence supporting a significant dose escalation to the GTV with focal boost techniques. However, the debate persists regarding achieving an escalated dose to the boost volume while sparing critical Organs at Risk (OARs). It was noted that higher doses to the boost volume and lower doses to the remaining prostate volume may not significantly affect OARs, especially in patients with favorable anatomy or where steep dose gradients can be sculpted.

The study also explored the potential benefits of combining focal boost with other treatment modalities like VMAT or brachytherapy to optimize dose delivery and minimize side effects. Overcoming challenges related to OAR overlap and achieving dose homogeneity in treatment planning was acknowledged as crucial.  Further, study suggested future developments involving the prediction of biological information and the use of MRI-guided radiation therapy to further enhance treatment outcomes in cancer care. Overall, while informative, the study could benefit from shorter sentences and more concise language. Redundant information repetition should be avoided, and some points could be combined to enhance readability.

Minor comments:

There are grammatical mistakes need to be corrected. Many long sentences. Authors are advised to break long sentence to short and easily readable ones.

The title has "trialse" which  should be "trials"

Suggestions for Rewriting:

·       Unsurprisingly, in all reviewed studies that included a plan comparison between conventional and focal boost radiotherapy, a significant dose escalation to the GTV was consistently achieved [45, 58, 76, 124]. However, it remains debatable whether both an escalated dose to the boost volume and improved OAR sparing can be achieved in the context of a focal boost. For plans that involved higher doses prescribed to the boost volume and lower doses prescribed to the rest of the prostate volume, similar or not significantly different dose to OARs were generally reported [12, 45, 65, 68, 76, 77, 124]. For example, significantly better protection to the bladder and rectum was reported with focal boost plans with dose de-escalation to the prostate (35.2 Gy to prostate PTV with 40 Gy to PTVboost in 5 fractions) compared with a 5 fractions whole-gland dose of 40 Gy (homogenous dose escalation to the entire gland) [70].

o   Comment: It's not clear what you're trying to convey in this paragraph. Consider breaking it down into smaller, more focused sentences to enhance readability and clarity.

There are many more such paragraphs that need to be rewritten for belter clarity for example see below:

Original text: The focal boost technique has the potential to be combined with other treatment techniques. Ciabatti et al. [59] compared focal boost VMAT plans with and without strict dose constraints applied to critical structures related to sexual function (including the penile bulb (PB), corpora cavernosa (CC), internal pudendal arteries (IPAs), and neurovascular bundles (NVB)) in a 12-patient cohort, by using the same dose prescription for prostate and boost volume. It was found that the sexual-sparing approach achieved significantly better sparing of sexual-function-related OARs without compromising dose coverage to the prostate and boost volume. By comparing focal boost brachytherapy plans with up to two additional HDR needles within the boost volume to those with a standard needle arrangement, a higher boost level to the boost volume was achieved, although it was accompanied by slightly higher doses to the rectum [61]. Instead of boosting the delineated IPL volume, Amini et al. [125] defined the biopsy positive prostate lobe as the boost volume. The doses delivered to the surrounding critical structures by this method were reported to be lower than the conventional prescription without the affected-lobe boost, and because a lower dose was delivered to the opposite NVB, the risk of erectile dysfunction (ED) was also expected to be lower after treatment.

Revised text : The focal boost technique can be combined with other treatment techniques. Ciabatti et al. [59] conducted a comparison between focal boost VMAT plans with and without strict dose constraints applied to critical structures related to sexual function, such as the penile bulb (PB), corpora cavernosa (CC), internal pudendal arteries (IPAs), and neurovascular bundles (NVB), in a 12-patient cohort. They used the same dose prescription for the prostate and boost volume and found that the sexual-sparing approach achieved significantly better sparing of sexual-function-related OARs without compromising dose coverage to the prostate and boost volume. Another study compared focal boost brachytherapy plans with up to two additional HDR needles within the boost volume to those with a standard needle arrangement, resulting in a higher boost level to the boost volume, albeit with slightly higher doses to the rectum [61]. In a different approach, Amini et al. [125] defined the biopsy-positive prostate lobe as the boost volume. This method led to lower doses delivered to the surrounding critical structures compared to the conventional prescription without boosting the affected lobe, reducing the risk of erectile dysfunction (ED) after treatment.

Decision:

Accept after these comments/concerns being addressed.

see above comments

Reviewer 4 Report

This work reviews 34 planning studies and 35 clinical trials published between 2013 and 2023; it tries to evaluate the benefits deriving form a focal boost raditherapy in terms of clinical outcome and local tumor control. Authors also considered the adverse effects of the focal boost therapy and reviewing the selected studies they found out that no significant differences are reported in toxicity between focal boost and conventional radiotherapy; authors conclude the review saying that focal boost therapy has the potential to be a new standard of care.

My suggestions:

A legend grouping all the acronyms would make the reading easier. Please add it

Not all the reviewed study use the same radiological technique to assess IPLs or to follow them up after the therapy. This could represent a bias

A randomized clinical trial with a standardized diagnostic and therapeutic protocol comparing focal boost therapy and conventional radiotherapy could give stronger evidence to the conclusions. New and more accurate diagnostic techniques are also available to streghten the evidences: at this regard I can suggest the analysis of this work: https://pubmed.ncbi.nlm.nih.gov/34046207/

Authors only took in exam cases of non-metastatic prostate cancer, but it’s universally accepted that also oligometastatic cancers can benefit from radiotherapy; so authors could enlarge the number of studies selected for the review, including also oligometastatic prostate cancers. At this regard I can kindly suggest the reading of this work: https://pubmed.ncbi.nlm.nih.gov/36010331/

Minor editing

Round 2

Reviewer 2 Report

I thank the authors to modify the text as I requested.

I keep in my position, that it will be useful, if they write also the Dose in Gy and not only in %

Author Response

Comment #1:

I thank the authors to modify the text as I requested.

I keep in my position, that it will be useful, if they write also the Dose in Gy and not only in %

Response #1:

Thanks for your comment. Figure 2 has been added at lines 330 to 333 according to your suggestions. The Figurte 2 shows the BEDs prescribed to prostate (PTV) and boost volume (GTV) in Gy.

Reviewer 4 Report

Authors have to update references https://pubmed.ncbi.nlm.nih.gov/36010331/

according to reviewer suggestions and according to changes

Minor Editing.

Author Response

Comments #1:

Authors have to update references https://pubmed.ncbi.nlm.nih.gov/36010331/

according to reviewer suggestions and according to changes

Response #1:

Your reference (reference 4) has been added at lines 52 to 54:

"Radiation therapy has firmly established itself as the primary treatment for PCa, with positive results in the treatment of both localized [3] and metastatic disease [4]."

Comments on the Quality of English Language:

Minor Editing.

Response to comments on the quality of English language:

Two of the co-authors are English first-language speakers and helped edit the manuscript. If revision is required, we would be grateful for examples of such revision.